# Lecturers' readiness for EMI in Malaysia higher education

**Yueh Yea Lo**⬥*, **Juliana Othman**

Department of Language and Literacy Education, Faculty of Education, University of Malaya, Kuala Lumpur, Malaysia

* janice@um.edu.my

## Abstract

The current study aims to examine lecturer readiness for English Medium Instruction (EMI) in higher educational institutions and the contextual influences of gender, age, academic qualification, teaching experience, EMI course teaching involvement, and EMI training. A quantitative research design was employed, and a survey questionnaire was completed by 227 lecturers (out of 250 invited participants) from private universities in Klang Valley, Malaysia to gauge self-ratings of personal knowledge, skills, abilities, and attitudes in educating EMI courses. The collected data were subsequently analysed via the Statistical Package for Social Sciences (SPSS) version 27.0 software before revealing the findings from the inferential statistics of the t-test and one-way analysis of variance (ANOVA) on lecturers' gender, age, academic qualification, teaching experience, EMI course teaching involvement, and EMI training. Resultantly, the important role of lecturers' knowledge, understanding, skills, abilities, and attitudes was highlighted to further enhance intercultural communicative competence in managing the increasingly diversified student body in EMI classrooms.

**Data Availability Statement:** All relevant data are within the manuscript and its supporting information.

**Funding:** The author(s) received no specific funding for this work.

## Introduction

The use of English as the medium of instruction (EMI) in higher education institutions is a growing phenomenon on a global scale [1, 2]. Numerous countries have demonstrated commitment and support toward the EMI by implementing various pertinent policies and programmes in the education systems [3–6]. For example, European nations integrate English and discipline-specific communication in universities by gradually replacing English for Specific Purposes (ESP). Meanwhile, the EMI remains an ambiguous [7] yet evolving term in educational research [8], as indicated by the director of the Oxford Centre for Research and Development on EMI, Ernesto Macaro, who explicitly stated that 'we do not yet know what EMI is' ([9], p. 5). Nonetheless, the general EMI definitions include four elements, namely "(i) English is used for instructional purposes, (ii) English is not itself the subject being taught, (iii) language development is not a primary intended outcome, and (iv) English is a second language (L2) for most participants in the setting ([10], p. 499). In Malaysia, English is the L2 for most citizens, which renders the EMI a source of contention in both standard and higher

**Competing interests:** The authors have declared that no competing interests exist.

educational institutions. Numerous public and private universities have either adopted or are in the process of adopting the EMI [11], which is in line with the Malaysian Ministry of Education (MOE) advocating for establishing the nation as a world-class education and research hub.

As the EMI is an inherent part of the language policy, Malaysia endeavours to resolve several challenges in developing effective intercultural communication across classrooms [12, 13]. Generally, the success of EMI programmes is contingent on lecturers who would determine the character with personal attitudes and opinions [14, 15]. Particularly, dissimilar attitudes and opinions would hinder the successful EMI implementation when lecturers possess limited knowledge, assistance, and support required for the related practices [16]. Therefore, lecturers' EMI readiness would be the primary factor in influencing the EMI implementation effectiveness. EMI readiness could be conceptualised as perceived knowledge, skills, and beliefs towards a high diversity of language practices across various EMI classrooms [17] regardless of the students' language proficiency or the permission to continue personal mother tongue languages until demonstrating sufficient EMI readiness. In the present study, higher education lecturers' EMI readiness refers to lecturers' knowledge, understanding, skills, abilities, and attitudes towards the EMI. Past studies consistently underscored the significance of lecturer EMI readiness in elevating intercultural communicative competence to facilitate the success of EMI programmes [18–20]. Nevertheless, more empirical evidence is imperative to delineate lecturer EMI readiness in developing nations, such as Malaysia.

The present study sought to bridge the existing literature gap by contributing additional theoretical knowledge regarding Malaysian lecturer EMI readiness in higher education institutions with the contextual influences of gender, age, academic qualification, teaching experience, EMI course teaching involvement, and EMI training. Accordingly, three research questions were formulated to guide the current study, namely, (1) How ready are lecturers in terms of knowledge, understanding, skills, abilities, and attitudes based on gender, age, academic qualification, and teaching experience?; (2) Do significant differences exist in lecturers' knowledge, understanding, skills, abilities, and attitudes based on gender, age, academic qualification, and teaching experience?; and (3) Do significant differences exist in lecturers' knowledge, understanding, skills, abilities, and attitudes from involvement in inculcating EMI courses and EMI training on EMI in higher education?. This article begins with a literature review on lecturer EMI readiness followed by descriptions of the research methodology. Subsequently, empirical findings are presented and discussed to emphasise lecturers' important role in further enhancing intercultural communicative competence to manage the increasingly diversified EMI classrooms filled with students of various nationalities, attitudes, linguistic backgrounds, levels of prior knowledge, and learning styles.

## Literature review

### The EMI readiness in higher education

Readiness is generally defined as the integration of knowledge, skills, and attitudes [21] as a prerequisite for adequate job functioning [22]. Existing theories on work readiness emphasise that employees should understand job scopes and acquire relevant abilities before effectively applying knowledge and skills with appropriate dispositions to achieve satisfactory levels of job competencies [22, 23]. As such, knowledge, skills, and attitudes in the same professional task are recommended to be measured simultaneously when the action pattern in job obligations is an emergent characteristic [24].

According to Monico et al. [25], lecturers are required to accomplish a balance of knowledge, skills, and attitudes, which would be instrumental to apply inclusive pedagogies and

maintaining a positive attitude towards higher educational EMI. Contemporarily, lecturers are required to inculcate students with diverse requirements and experiences [17], although most lecturers lack the necessary skills and readiness to implement EMI programmes while fulfilling diverse student needs [26]. Thus, lecturer readiness is an important factor in successful higher education EMI implementation.

Previous researchers revealed that lecturers who were unprepared to satisfy contemporary students' diverse demands, responsibilities, needs, and experiences would not favour the EMI [27–29] owing to high unfamiliarity with the higher education EMI requirements, practices, required resources, and additional workload [30–32]. The lack of preparation would consequently engender the inability of higher educational institutions to execute various EMI programmes effectively in international classrooms [20]. For instance, Dearden [1] unveiled that lecturer were aware of the higher education EMI concept but were uncertain of the implementation approaches in diversified EMI classrooms. Subsequently, Dearden [1] appealed to reconstruct existing EMI training programmes to equip lecturers with pertinent linguistic, pedagogical, and intercultural skills before inculcating EMI students with diverse needs and experiences. Contrarily, lecturers who are inadequately trained with appropriate EMI teaching strategies would exhibit a negative attitude towards international students, thereby lessening the success likelihood of conducting EMI courses [33]. Therefore, higher education EMI success is closely related to lecturer acceptance and recognition of linguistically and culturally diverse students [34] through a wider teaching repertoire, such as designing productive instructions to support and accommodate student learning difficulties [35].

## Defining EMI

The term EMI is unclear, even among researchers. EMI can be applied broadly, embracing any teaching strategies incorporating content in a second language. Table 1 demonstrates the variety of EMI definitions used in the scientific literature.

According to definitions of EMI found in the literature (see Table 1), its emphasis on subject-content mastery sets it apart. Each of the interpretations shown in Table 1 highlights the value of academic content and the lack of overt language acquisition goals in EMI courses. Nonetheless, this does not imply that EMI courses cannot be designed to help improve students' English skills. The many definitions of EMI emphasise that content mastery is just one of EMI's primary goals—not its sole one.

Given the complexity of the EMI phenomenon and the range of contexts in which EMI is being introduced and developed, the concept itself is challenging to define. As a result, this study adopts the definition of EMI given below, aware that some of its elements may be contested:

**Table 1. Variety of EMI definitions in the literature.**

| Year | EMI definitions |
|------|-----------------|
| 2011 | "The central focus is on students' content mastery and no language aims are specified" (Unterberger & Wilhelmer [36], p. 96) |
| 2012 | "focuses on content learning only" (Smit & Dafouz [37], p. 4) |
| 2014 | "English-taught degree programs. . . predominately aim at the acquisition of subject knowledge" (Unterberger [38], p. 37) |
| 2014 | "the use of English to teach academic subjects in countries or jurisdictions where the first language (L1) of the majority of the population is not English" (Dearden [1], p. 4) |
| 2016 | "an umbrella term for academic subjects taught through English, one making no direct reference to the aim of improving students' English" (Dearden & Macaro [32], p. 456) |

The use of English language to teach academic subjects (other than English itself) in countries or jurisdictions where the first language of the majority of the population is not English. ([1], p.4)

The idea of "academic subjects (other than English itself)" is one particular aspect of the concept on which this study focuses. As such, it is helpful to review any prior study that identified the disciplines that use EMI the most. According to recent research by Macaro et al. [17] that looked at a data set of 83 EMI studies, they found that engineering (n = 23) and business administration and management (n = 21) were the two disciplines that used EMI most frequently. However, it should be noted that some EMI studies gave broad categories of disciplines such as the sciences, social sciences, and humanities. This means that EMI usage in the two fields described above may not be entirely reflective of EMI in practice.

## Lecturers' knowledge and understanding

Prior research consistently associated lecturers' knowledge and understanding with personal EMI readiness in higher education [27, 39–41]. For example, Dang, Bonar, and Yao [42] asserted that lecturers would be required to undergo intensive training in equipping the lecturers with sufficient EMI readiness, while Airey [7] affirmed that EMI knowledge would significantly contribute to effective EMI practices in higher educational institutions. In addition, Fenton-Smith, Stillwell, and Dupuy [27] emphasised the importance of lecturer training curricula, especially in accounting for students with diverse linguistic needs and experiences, wherein thoroughly understanding EMI practices should be a mandatory course in lecturer training and professional development. Macaro and Han [41] also proposed that fundamental knowledge, such as recognising students' fundamental abilities, and pedagogical skills, including instructional differentiation, would be highly vital in effectively conducting an EMI classroom. Meanwhile, Floris [40] postulated that the ESP knowledge fine-tuned to resolve linguistic issues in EMI classrooms would be particularly crucial for lecturers to address multitudinous learning needs.

## Lecturers' skills and abilities

Lecturer EMI readiness in higher education was manifested by past scholars to be highly correlated to lecturers' skills and abilities [42–44]. Specifically, lecturers' capacities to perform tasks independently, attend to obligations efficiently, adhere to instructions accurately, and communicate with peers effectively would be essential to assist and prepare students with contrasting linguistic requirements and backgrounds in the EMI transition [45]. Dang, Bonar, and Yao [42] highlighted the lecturers' role to improve personal knowledge and skills through formal and informal training, such as seminars and workshops. Nonetheless, O' Dowd [44] discovered that EMI training programmes did not adequately encompass the required knowledge and skills for lecturers to effectively inculcate students with various learning needs, which could render high uncertainty and unconfidence to perform adaptations in relevant teaching instructions [46–48]. Summarily, lecturers who were sufficiently trained in the higher education EMI would possess higher have higher efficacy than their inadequately trained counterparts to conduct EMI courses [26, 27].

## Lecturers' attitudes

Lecturers' attitudes have emerged as a significant factor in successful higher education EMI implementation [27, 49] although existing literature demonstrated mixed results. Several studies discovered that lecturers who exhibited positive attitudes towards the higher education EMI [2, 50], while other academicians revealed a neutral or negative attitude [51, 52]. On the one hand, lecturers who possess experience and exposure to educating students with

heterogeneous needs and experiences would be more tolerant and open to EMI programmes [53, 54]. Similarly, Jensen and Thogersen [28] discovered that younger Danish university lecturers who conducted teaching mainly in English manifested more positive views towards the EMI compared to their older equivalents with long teaching experience but limited EMI exposure and training. Contrarily, Macaro and Akincioglu [55] disclosed a significant decline in lecturers' positive attitudes toward the EMI throughout the teaching years. As such, further studies are recommended to determine the relationship between lecturers' attitudes and EMI acceptance based on gender, age, academic qualifications, teaching experience, EMI course teaching involvement, and EMI training.

## Methodology

### Research methods

A quantitative methodology was employed through survey dissemination to collect data regarding lecturer EMI readiness in Malaysian higher education. Participants were recruited purposively before subsequently distributing 250 questionnaires in person during the final semester meeting between January and February 2022. Resultantly, only 227 lecturers, who were the survey respondents, completed the questionnaire with a response rate of 90.8%. The participation was voluntary and the participants' data or responses were assured not to fall into the hands of a third party to ensure response confidentiality.

### Research participants

The survey participants were 227 lecturers recruited from private universities in Klang Valley, Malaysia. Amongst the 227 lecturers, 159 were women (70.04%) and 68 were men (29.96%) from four different disciplines in social sciences and humanities, namely education, linguistics, philosophy, and psychology (see Table 2). The age-wise distributions were 18.06%, 23.79%, 25.99%, and 32.16% respectively in the age range between 25 and 29 years old, 30 and 34 years old, 35 and 39 years old, and above 40 years old. Additionally, 58.59%and 41.41% of the participants obtained academic qualifications of Master's degree and doctoral degree respectively. The number of teaching years range from one to 20, which were 18.06%, 23.79%, 32.16%, and 25.99% respectively for under five years, 6 to 10 years, 11 to 15 years, and above 15 years. The number of lecturers who previously attended the higher education EMI training was 29.96% versus 70.04% of the counterparts who did not. Table 2 below shows more detailed information regarding participants' demographic data.

### Research instrument

The study questionnaire was adapted from Bolton and Kuteeva's [56] study, which explored the English issue at multiple instruction levels encountered by Swedish universities. The questionnaire items were considered carefully through pertinent refinements during several development stages in October 2021. The refinement was conducted by reviewing existing literature and soliciting feedback from experts. The pilot study was conducted in December 2021 with six lecturers from two private universities in Klang Valley, Malaysia. Two items were revised for enhanced meaning clarity. For instance, the item "I understand the purpose for EMI" was amended to "I understand the purpose of EMI in higher education" and the item "I am able to explain the course material well" to "I am able to explain the course material through EMI to students with diverse needs and experiences."

The questionnaire consisted of two sections, wherein the first section comprised demographic variables, including gender, age, academic qualifications, and teaching experience.

**Table 2. Demographic data of participants.**

| Category | Options | No. of participants | Percentage (%) |
|---|---|---|---|
| Discipline | Education | 66 | 29.07 |
| | Linguistics | 55 | 24.23 |
| | Philosophy | 49 | 21.59 |
| | Psychology | 57 | 25.11 |
| Gender | Male | 68 | 29.96 |
| | Female | 159 | 70.04 |
| Age | 25–29 | 41 | 18.06 |
| | 30–34 | 54 | 23.79 |
| | 35–39 | 59 | 25.99 |
| | Above 40 | 73 | 32.16 |
| Academic Qualifications | Master's degree | 133 | 58.59 |
| | Doctoral degree | 94 | 41.41 |
| Years of Teaching Experience | < 5 years | 41 | 18.06 |
| | 6–10 years | 54 | 23.79 |
| | 11–15 years | 73 | 32.16 |
| | Above 15 years | 59 | 25.99 |
| Participated in EMI Training | Yes | 68 | 29.96 |
| | No | 159 | 70.04 |

Furthermore, the section also contained questions regarding EMI training, involvement in teaching EMI courses, and exposure to the EMI in higher education, such as relevant concise courses. The second section was composed of 24 items categorised into three dimensions. The first dimension "knowledge and understanding" contained eight items, such as "I understand the purpose of EMI in higher education". The second dimension "skills and abilities" encompassed 8 items, including "I am able to teach through EMI to students with diverse needs and experiences". The third dimension "attitudes" included 8 items, for example, "I care for the progressive learning of students with diverse needs and experience in the EMI classroom". A 4-point Likert Scale was applied to every questionnaire item, which ranged from 1 (Strongly Disagree) to 4 (Strongly Agree), in measuring lecturer higher education EMI readiness.

## Data analysis

The data were analysed by administering the Statistical Package for Social Sciences (SPSS) 27.0. software. Descriptive statistics of mean and standard deviation were generated to fulfil the first research question. Meanwhile, the second and third research questions were answered with the inferential statistics of the t-test, and one-way analysis of variance (ANOVA).

## Reliability

The reliability degrees of the three dimensions were assessed via Cronbach's alpha as portrayed in Table 3. The results demonstrated that the internal consistency reliability of the three dimensions ranged from acceptable to good levels.

**Table 3. Reliability of the instrument.**

| Dimensions | No. of items | Cronbach alpha |
|---|---|---|
| Knowledge and understanding | 8 | .86 |
| Skills and abilities | 8 | .89 |
| Attitudes | 8 | .86 |

## Ethical considerations

The Research and Governance Framework of that institution served as the framework for this study. While the researchers recognised the need for confidentiality and informed consent, the researchers realised that the framework for ethical requirements for studies of this nature is likely to differ in different universities and faculties. For this reason, the description of the ethical protocols is therefore essential and inevitable. The ethical norms used in this article were derived from two stages.

In the first stage, authorisation was proactively sought from the appropriate gatekeepers. For this study, which examines the lecturers' readiness for EMI in Malaysian higher education from two private universities in Klang Valley, Malaysia, it is imperative to obtain approval from the heads of departments before inviting all the lecturers in the pertinent departments for this study. After obtaining written permission to invite lecturers and carry out this survey research, the second stage was initiated.

The second stage was undertaken within the study itself. A face-to-face verbal explanation for participants was accompanied by written information with the opportunity to raise further questions and concerns. Before data collection, the process of obtaining participants' written consent includes (i) giving the participants options to participate or decline and (ii) emphasising the ethical practices for privacy, anonymity, and confidentiality while collecting, analysing, and reporting data.

## Results

To measure lecturer readiness levels in terms of knowledge, understanding, skills, abilities, and attitudes towards the EMI in higher education (the first research question), the means of all items and the overall mean of each measurement dimension were computed. The item means were also compared based on the ranking of the low, medium, and high levels, which were derived by dividing the range of scores into three equal intervals, namely 1.00–1.99 (low), 2.00–2.99 (moderate), and 3.00–4.00 (high). Table 4 depicts the means and respective rankings of the dimension "knowledge and understanding". Accordingly, the mean score of 2.909 suggested that lecturers' higher education EMI knowledge and understanding were at a moderate level. Five items under the dimension were also ranked moderate with mean scores ranging from 2.607 to 2.869. The lowest ranked item was "I have knowledge of assessing students with diverse needs and experiences in the EMI classroom". Apart from the moderately ranked items, the remaining three items were ranked high with mean scores between 3.192 to 3.204. The highest ranked item within this category states that "I understand the meaning of EMI in higher education". Summarily, the findings postulated that although lecturers possessed a relatively high understanding level of broader EMI aspects in higher education, a deeper

**Table 4. Level of lecturers' knowledge and understanding about EMI in higher education (n = 227).**

| Items No. | Item | Mean | SD | Rank |
|---|---|---|---|---|
| 1 | I understand the meaning of EMI in higher education | 3.204 | 0.694 | High |
| 2 | I understand the purpose of EMI in higher education | 3.192 | 0.732 | High |
| 3 | I understand the process involved in teaching academic subjects through EMI | 3.192 | 0.732 | High |
| 6 | I have knowledge of how to teach students with diverse needs and experiences in the EMI classroom | 2.869 | 0.478 | Moderate |
| 4 | I understand the university curriculums and policies on EMI in higher education | 2.843 | 0.690 | Moderate |
| 5 | I have the knowledge to create a conducive learning environment in an EMI classroom | 2.757 | 0.722 | Moderate |
| 7 | I have the knowledge to sustain the learning in the EMI classroom | 2.608 | 0.721 | Moderate |
| 8 | I have knowledge of assessing students with diverse needs and experiences in the EMI classroom | 2.607 | 0.669 | Moderate |

**Table 5. Level of lecturers' skills and abilities pertaining to EMI (n = 227).**

| Items No. | Item | Mean | SD | Rank |
|---|---|---|---|---|
| 1 | I am able to teach students with diverse needs and experiences through EMI | 2.874 | 0.713 | Moderate |
| 2 | I am able to explain the course material well to students with diverse needs and experiences through EMI | 2.845 | 0.689 | Moderate |
| 6 | It is hard to control students with diverse needs and experience in EMI classroom | 2.829 | 0.648 | Moderate |
| 5 | I am able to discuss with students the strategies needed to follow the lecture delivered through EMI | 2.763 | 0.665 | Moderate |
| 4 | I am able to discuss with students regarding the English-language-related challenges through EMI | 2.573 | 0.741 | Moderate |
| 3 | I am able to discuss with students the subject issues at length with relative ease and accuracy through EMI | 1.940 | 0.771 | Low |
| 7 | I need support from English language lecturers if I have to teach students with diverse needs and experiences through EMI | 1.924 | 0.749 | Low |
| 8 | I need extra effort to teach students with diverse needs and experience through EMI | 1.915 | 0.720 | Low |

knowledge of specific aspects, such as assessments, modifications to learning in a specific context, and identifications of the types of students' EMI needs, is required as it plays a significant role in the teaching and assessment processes.

Table 5 demonstrates lecturers' EMI skills and abilities in higher education at a moderate level with a mean score of 2.458. Five items under the dimension "skills and abilities" achieved a moderate mean score ranging from 2.573 to 2.874. The highest ranked item was "I am able to teach students with diverse needs and experiences through EMI". Contrastingly, three items were situated at a low level with mean scores spanning from 1.915–1.940. The lowest ranked item was "I need extra effort to teach students with diverse needs and experiences through EMI". Summarily, no items attained a high level, which suggested more effort and support were required by the lecturers to conduct the EMI courses, albeit with satisfactory levels of skills and abilities.

Table 6 highlights the mean score of the dimension "attitudes" of lecturers towards higher education EMI. The results revealed that lecturers' EMI attitudes were at a high level with a mean score of 3.002. Six items under this dimension were also at a high level with a mean score ranging between 3.000 to 3.194. The highest ranked item was "I believe students with diverse needs and experiences in the EMI classroom can achieve their best with support". The remaining two items were ranked at a moderate level with a mean score ranging from 2.636 to 2.688. Despite lecturers' generally positive EMI attitudes, managing students with diverse learning needs and experiences remained a significant challenge.

The second research question guided the discovery of a significant difference in lecturers' (a) knowledge and understanding, (b) skills and abilities, and (c) attitudes concerning several independent variables, which were gender, age, academic qualification, and teaching

**Table 6. Level of lecturers' attitudes towards EMI in higher education (n = 227).**

| Items No. | Item | Mean | SD | Rank |
|---|---|---|---|---|
| 4 | I believe students with diverse needs and experiences in the EMI classroom can achieve their best with support | 3.194 | 0.671 | High |
| 3 | I care for the skill advancement of students with diverse needs and experiences in the EMI classroom | 3.192 | 0.680 | High |
| 2 | I care for the achievement of students with diverse needs and experiences in the EMI classroom | 3.145 | 0.664 | High |
| 6 | I believe students with diverse needs and experiences in the EMI classroom can adopt learning strategies to compensate for comprehension problems with support | 3.082 | 0.732 | High |
| 5 | I believe students with diverse needs and experiences in the EMI classroom can be equipped with both English and subject knowledge | 3.082 | 0.732 | High |
| 1 | I care for the progressive learning of students with diverse needs and experiences in the EMI classroom | 3.000 | 0.772 | High |
| 7 | Students with diverse needs and experiences will interrupt the teaching and learning process through EMI | 2.688 | 0.702 | Moderate |
| 8 | I do not feel confident and comfortable teaching academic subjects through EMI | 2.636 | 0.739 | Moderate |

**Table 7. Difference in lecturer readiness by gender (n = 227).**

| Dimensions | Gender | n | Mean | SD | Sig. (2-tailed) |
|---|---|---|---|---|---|
| Knowledge and understanding | Female | 159 | 32.097 | 4.979 | .005* |
| | Male | 68 | 31.053 | 5.037 | |
| Skills and abilities | Female | 159 | 21.553 | 3.775 | .085 |
| | Male | 68 | 21.530 | 3.526 | |
| Attitudes | Female | 159 | 20.996 | 2.701 | .630 |
| | Male | 68 | 21.306 | 3.103 | |

experience. Accordingly, the independent sample t-test was administered to examine whether a significant difference existed in the three dependent variables concerning gender. The findings revealed a significant difference in lecturers' EMI knowledge and understanding in higher education between females ($M = 32.097$, $SD = 4.979$) and males ($M = 31.053$, $SD = 5.037$), with a mean difference of 1.044 ($p = .005$; see Table 7). Nevertheless, no significant differences were discovered in lecturers' skills and abilities ($ss = .085$) and attitudes ($ss = .630$) between genders.

The one-way ANOVA was employed to examine whether a significant difference emerged in lecturers' (a) knowledge and understanding, (b) skills and abilities, and (c) attitudes based on age in four categories, namely (i) between 25 to 29 years old, (ii) between 30 and 34, (iii) between 35 to 39, and (iv) above 40. The results (see Table 8) manifested an insignificant difference in lecturers' higher education EMI knowledge and understanding between the four age groups ($p = .612$). Similarly, an insignificant difference was demonstrated in lecturers' skills and abilities between the four age groups ($p = .595$). Contrarily, a significant difference was uncovered in lecturers' attitudes between the four age groups ($p = .005$), hence suggesting that elder lecturers would be inclined to exhibit a higher degree of negative attitudes towards higher education EMI.

The one-way ANOVA was also conducted to determine the difference in lecturers' (a) knowledge and understanding, (b) skills and abilities, and (c) attitudes based on respective academic qualifications at two levels, which were (i) Master's degree and (ii) doctoral degree. The findings (see Table 9) uncovered a significant difference in lecturers' EMI knowledge and understanding between the Master's degree ($M = 30.931$, $SD = 5.267$) and Doctoral degree ($M = 32.090$, $SD = 5.625$), with a mean difference of 1.159 ($p = .002$). A significant difference was also manifested in lecturers' EMI skills and abilities between the Master's degree

**Table 8. Difference in lecturer readiness by age (n = 227).**

| Dimensions | Age | n | Mean | SD | Sig. (2-tailed) |
|---|---|---|---|---|---|
| Knowledge and understanding | 25–29 | 41 | 21.086 | 2.840 | .612 |
| | 30–34 | 54 | 21.599 | 3.245 | |
| | 35–39 | 59 | 21.306 | 3.103 | |
| | > 40 | 73 | 21.181 | 2.949 | |
| Skills and abilities | 25–29 | 41 | 20.996 | 2.701 | .595 |
| | 30–34 | 54 | 21.409 | 3.166 | |
| | 35–39 | 59 | 21.205 | 3.013 | |
| | > 40 | 73 | 20.918 | 2.619 | |
| Attitudes | 25–29 | 41 | 31.718 | 5.496 | .005* |
| | 30–34 | 54 | 31.500 | 5.302 | |
| | 35–39 | 59 | 31.051 | 5.035 | |
| | > 40 | 73 | 30.727 | 5.185 | |

**Table 9. Difference in lecturer readiness by academic qualification (n = 227).**

| Dimensions | Academic qualification | n | Mean | SD | Sig. (2-tailed) |
|---|---|---|---|---|---|
| Knowledge and understanding | Master's degree | 133 | 30.931 | 5.267 | .002* |
| | Doctoral degree | 94 | 32.090 | 5.625 | |
| Skills and abilities | Master's degree | 133 | 30.767 | 4.968 | .005* |
| | Doctoral degree | 94 | 32.108 | 4.887 | |
| Attitudes | Master's degree | 133 | 21.409 | 3.166 | .546 |
| | Doctoral degree | 94 | 21.691 | 3.092 | |

($M = 30.767$, $SD = 4.968$) and the Doctoral degree ($M = 32.108$, $SD = 4.887$; $p = .005$). The results indicated that lecturers with higher qualification levels would demonstrate higher levels of EMI knowledge and skills in higher education. Conversely, an insignificant difference in lecturers' attitudes between the two groups ($p = .546$).

The one-way ANOVA was administered to assess the difference in lecturers' (a) knowledge and understanding, (b) skills and abilities, and (c) attitudes based on teaching years into four categories, namely (i) under 5 years, (ii) between 6 and 10 years, (iii) between 11 and 15, and (iv) above 15. The results (see Table 10) demonstrated insignificant differences in lecturers' higher education EMI knowledge and understanding ($p = .544$) and skills and abilities ($p = .449$) between the four categories. Contrastingly, a significant difference was revealed in lecturers' EMI attitudes between the four groups ($p = .005$), thus postulating that experienced lecturers might manifest a more negative attitude towards EMI in higher education.

To address the third research question, the existence of a significant difference in lecturers' (a) knowledge and understanding, (b) skills and abilities, and (c) attitudes were analysed based on (i) involvement in teaching EMI courses and (ii) exposure to EMI training in higher education. To facilitate the relevant analysis, lecturers' EMI teaching experience was dichotomised, wherein one group possessed under one year of teaching experience while another possessed above one year of experience. Table 11 delineates the independent samples t-test results, in which an insignificant difference in lecturers' EMI knowledge and understanding was demonstrated between lecturers with minimal EMI teaching experience ($M = 32.090$, $SD = 5.625$) and their equivalents with above one year of EMI inculcation experience ($M = 31.277$, $SD = 4.832$; $p = .447$). Similarly, an insignificant difference was uncovered in lecturers' attitudes between the minimally experienced group ($M = 31.363$, $SD = 4.829$) and the more experienced

**Table 10. Difference in lecturer readiness by years of teaching experience (n = 227).**

| Dimensions | Teaching experience (yrs.) | n | Mean | SD | Sig. (2-tailed) |
|---|---|---|---|---|---|
| Knowledge and understanding | < 5 | 41 | 31.779 | 4.912 | .544 |
| | 6–10 | 54 | 31.823 | 5.636 | |
| | 11–15 | 73 | 31.793 | 5.117 | |
| | > 15 | 59 | 31.051 | 5.035 | |
| Skills and abilities | < 5 | 41 | 32.090 | 5.625 | .449 |
| | 6–10 | 54 | 31.421 | 5.446 | |
| | 11–15 | 73 | 31.404 | 5.830 | |
| | > 15 | 59 | 31.363 | 4.829 | |
| Attitudes | < 5 | 41 | 31.718 | 5.496 | .005* |
| | 6–10 | 54 | 31.500 | 5.302 | |
| | 11–15 | 73 | 30.727 | 5.185 | |
| | > 15 | 59 | 31.051 | 5.035 | |

**Table 11. Difference in lecturer readiness by experience in teaching students through EMI (n = 227).**

| Dimensions | Experience in teaching through EMI | n | Mean | SD | Sig. (2-tailed) |
|---|---|---|---|---|---|
| Knowledge and understanding | 1 year or less | 136 | 21.090 | 5.625 | .427 |
| | More than 1 year | 91 | 21.277 | 4.832 | |
| Skills and abilities | 1 year or less | 136 | 30.767 | 4.968 | .005* |
| | More than 1 year | 91 | 32.108 | 4.887 | |
| Attitudes | 1 year or less | 136 | 21.363 | 4.829 | .524 |
| | More than 1 year | 91 | 21.530 | 4.775 | |

counterparts (*M* = 31.530, *SD* = 4.775; *p* = .546). Nonetheless, a significant difference was discovered in lecturers' skills and abilities between lecturers possessing minimal EMI inculcation experience (*M* = 30.767, *SD* = 4.968) and lecturers with higher degrees of EMI teaching experience (*M* = 32.108, *SD* = 4.887; *p* = .005).

Table 12 portrays the independent sample t-test results on the significant difference concerning personal (a) knowledge and understanding, (b) skills and abilities, and (c) attitudes between lecturers who had attended relevant EMI training versus their counterparts who had not. The findings unveiled a significant difference in lecturers' EMI knowledge and understanding of higher education between lecturers who were trained in the EMI (*M* = 32.671, *SD* = 4.674) and lecturers who were not (*M* = 32.447, *SD* = 4.769), with a small mean difference of 0.224 (*p* = .005). Similarly, a significant difference was revealed in lecturers' skills and abilities between EMI-trained lecturers (*M* = 33.137, *SD* = 5.549) and lecturers who had not attended EMI training (*M* = 32.971, *SD* = 5.257), with a small mean difference of 0.166 (*p* = .005). Furthermore, a significant difference was demonstrated between lecturers who were trained in EMI (*M* = 31.151, *SD* = 5.135) and their equivalents who were not (*M* = 30.907, *SD* = 5.037) in personal attitudes towards EMI in higher education, with a small mean difference of 0.244 (*p* = .005).

## Discussion

The present study aimed to investigate lecturer EMI readiness in Malaysian higher education in terms of (1) knowledge and understanding, (2) skills and abilities, and (3) attitudes. The results postulated that knowledge was crucial to developing lecturers' EMI instructional competency [41, 57] as demonstrated by the findings at a moderate level. Hence, the lecturers possessed a satisfactory level of understanding of the meaning and process involved in inculcating academic subjects through the EMI. Nevertheless, closer scrutiny of the results discovered that lecturers possessed a higher level of philosophical EMI knowledge, including the underlying meaning, purpose, and process, compared to their technical knowledge of concept implementation, such as creating and sustaining the EMI learning environment for students with contrasting learning requirements and experiences. Moreover, academicians also advocated the

**Table 12. Difference in lecturer readiness by EMI training (n = 227).**

| Dimensions | EMI training | n | Mean | SD | Sig. (2-tailed) |
|---|---|---|---|---|---|
| Knowledge and understanding | Yes | 68 | 32.671 | 4.674 | .005* |
| | No | 159 | 32.447 | 4.769 | |
| Skills and abilities | Yes | 68 | 33.137 | 5.549 | .005* |
| | No | 159 | 32.971 | 5.257 | |
| Attitudes | Yes | 68 | 31.151 | 5.135 | .005* |
| | No | 159 | 30.907 | 5.037 | |

investigation of lecturers' abilities to effectively improvise and accurately express EMI subject matters to the students while appraising the influence of diverse cultures on EMI learning effectiveness [58, 59].

The difference in lecturers' EMI knowledge was manifested as insignificant by EMI teaching years, which propounded that lecturers' knowledge was not significantly affected by personal EMI inculcation experience. Contrarily, significant differences would emerge in lecturers' EMI knowledge and understanding only from respective academic qualifications and exposure to higher education EMI training. The findings posited that lecturers would only acquire a significant amount of EMI knowledge and understanding from a certain number of attended EMI training programmes while affirming that pertinent EMI training could elucidate effective content delivery methods through English to lecturers in becoming more competent and prepared [59–61]. Concurrently, the findings also demonstrated the importance of lecturers' professional and pedagogical training to enhance their EMI knowledge and understanding in higher education. Nonetheless, Fenton-Smith, Stillwell, and Dupuy ([27], p. 197) opined that professional development programmes, such as "short-term training in an overseas Anglophone locale" would not be highly effective in enhancing lecturer competence, compared to holistic training programmes with systematic professional development in producing a sustainable knowledge of the higher education EMI implementation [33, 62, 63].

The present findings revealed that Malaysian lecturers in higher educational institutions possessed a moderate level of skills and abilities to educate EMI courses, whereas self-reported lecturers' competence in conducting EMI courses was comparatively lower. As such, EMI lecturers required comprehensive guidance and training to enhance personal skills in accommodating students' linguistic proficiency and development needs. Furthermore, lecturers' training and experience in teaching students through the EMI exhibited a significant difference in respective EMI skills and abilities, contrary to lecturers' age, gender, and general inculcation experience which did not uncover significant differences. The current results were consistent with past researchers [27, 63] who unveiled higher EMI efficacy of lecturers who were trained previously than their counterparts who were not. The findings thus postulated that lecturers without adequate EMI training and preparation would be less knowledgeable, skilful, and responsive to students' different learning environments in content knowledge and the English language [64]. This further suggests that having adequate EMI knowledge helps lecturers be sensitive in terms of their capacity to appropriately respond to the particular interaction and scenarios in English.

The current results also revealed that lecturers' attitudes towards the EMI were at a high level, apart from the influences of age, EMI training, and teaching experience. Simultaneously, when investigating the EMI research evolution in European higher educational institutions, Molino et al. [65] uncovered those factors including gender, EMI training, and inculcation experience, were significantly associated with lecturers' attitudes. Similar results were also discovered by Dearden and Macaro [32], in which higher education lecturers who possessed more EMI teaching experience and understanding were highly confident and open to various student-related issues. Jensen and Thogersen [28] also manifested that Denmark university lecturers' English proficiency and self-efficacy levels were highly associated with personal EMI attitudes. Nevertheless, queries remained on the appropriateness of selecting lecturers through EMI proficiency tests, as the procedure might pose consequential disadvantages to certain academic groups [66]. Owing to the current uncertainty, further investigation would be imperative to disclose more evidence to support a particular decision. Accordingly, a follow-up intervention of offering internal faculty/professional development on EMI and resource assistance utilising this same survey for private and public institutions may be appropriate.

## Conclusion

The current study aimed to assess lecturer EMI readiness in Malaysian higher education in terms of knowledge, understanding, skills, abilities, and attitudes. The study outcomes demonstrated lecturers possessed a moderate level of EMI knowledge, understanding, skills, and abilities. Simultaneously, differences existed in lecturer readiness based on gender, age, academic qualification, teaching experience, EMI course teaching involvement, and EMI training. The study has significant implications for stakeholders in Malaysian universities and other emerging EMI contexts because it reveals how cultural communicative competence may be effectively integrated as a vital learning and teaching tool within specific EMI disciplinary curricula. The data presented here emphasises the dangers of using 'one-size-fits-all' approaches to language policy in diverse multilingual EMI contexts. Since English is a second language in Malaysia, the Malaysian Ministry of Education ought to make policy decisions regarding lecturer readiness for EMI through training or include EMI trained individuals in their selection criteria to impart better quality education and enhance cultural communicative competence in EMI classrooms. Generally, the findings were within research expectations as numerous contemporary lecturers did not receive sufficient higher education EMI training and Malaysia was encountering several challenges in increasing the EMI lecturers' awareness that language was fundamental to effective and efficient content delivery. Furthermore, the study results underscored the EMI lecturers' important role in further enhancing personal linguistic and pedagogical skills for non-native English-speaking students. Lecturers who attended adequate higher education EMI training would be thoroughly prepared for being proficient and sensitive to students' multiple content knowledge and English language acquisition requirements through relevant EMI practices. Based on the study findings, two recommendations can be made, which are intended to serve as a foundation for any EMI lecturer training and continuing professional development at Malaysian higher education institutions. First, the EMI lecturer training programs in applied sciences, social sciences, and humanities should be more context-bound and needs-based, taking into account the perspectives of EMI lecturers as insiders. Second, EMI training programs should have two key components: pedagogical and linguistic training, with the former focusing on teaching techniques, strategies, and classroom management and the latter on language-related issues. To improve lecturers' reflection on their own classroom delivery ad practices, pedagogical training should use a variety of strategies (e.g., peer observation, coaching, and mentoring between senior and younger EMI lecturers), while linguistic training should focus on accuracy and intelligibility. Finally, perhaps accreditation and certification for such training programmes can be provided to further encourage EMI lecturers to engage in continuing professional development. Nevertheless, the current study was limited to two private higher institutions in Malaysia and the employment of a self-reported questionnaire, which potentially produced social desirability response bias. Correspondingly, the study findings were not utterly representative of the entire Malaysian lecturer population. Due to the increasing advocacy for elevating EMI exposure in higher education, future studies could adopt a qualitative methodology to extensively assess lecturer EMI readiness. Likewise, it would be beneficial to explore lecturers' readiness for EMI throughout Malaysia to get a better conclusion of having trained lecturers with a positive attitude toward EMI knowledge and training.

## Supporting information

**S1 Raw data.**
(DOCX)

**S1 File. Permission to inviting lecturers to participate in survey research.**
(DOCX)

**S2 File. Consent for participation in survey research.**
(DOCX)

## Acknowledgments

The reviewers have provided us with very insightful comments to improve our article. We gratefully acknowledge their thoroughness and generous feedback.

## Author Contributions

**Conceptualization:** Yueh Yea Lo.

**Formal analysis:** Yueh Yea Lo, Juliana Othman.

**Methodology:** Yueh Yea Lo.

**Validation:** Yueh Yea Lo, Juliana Othman.

**Writing – original draft:** Yueh Yea Lo.

**Writing – review & editing:** Yueh Yea Lo.

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
