## [Decision Letter · Decision Letter 0]

25 Jan 2023

PONE-D-22-27417Lecturers' readiness for EMI in Malaysia higher education: Fantasy or realities?PLOS ONE

Dear Dr. Yueh Yea Lo,

Thank you for submitting your manuscript to PLOS ONE. After careful consideration, we feel that it has merit but does not fully meet PLOS ONE’s publication criteria as it currently stands. Therefore, we invite you to submit a revised version of the manuscript that addresses the points raised during the review process.

ACADEMIC EDITOR:

Authors are requested to reply the queries/ suggestions, asked by both the reviewers.

We look forward to receiving your revised manuscript.

Kind regards,

Priti Chaudhary, M.S.

Academic Editor

PLOS ONE

and https://journals.plos.org/plosone/s/file?id=ba62/PLOSOne_formatting_sample_title_authors_affiliations.pdf.

Reviewers' comments:

Reviewer's Responses to Questions

**Comments to the Author**

1. Is the manuscript technically sound, and do the data support the conclusions?

Reviewer #1: Yes

Reviewer #2: Partly

2. Has the statistical analysis been performed appropriately and rigorously? 

Reviewer #1: Yes

Reviewer #2: Yes

3. Have the authors made all data underlying the findings in their manuscript fully available?

Reviewer #1: Yes

Reviewer #2: Yes

4. Is the manuscript presented in an intelligible fashion and written in standard English?

Reviewer #1: Yes

Reviewer #2: Yes

5. Review Comments to the Author

Reviewer #1: There is moderate level of role of lecturers readiness for EMI regarding knowledge,understanding,skills,abilities and attitudes .Since English is second language in Malaysia. Malaysian Ministry of education has to make policy decision for lecturers readiness for EMI by undergoing training or their selection criterion should have EMI trained persons to impart better quality of education to enhance cultural communicative competence in many increasingly diversified student body in EMI classroom.Difference existed in lecturers readiness based on gender age academic qualification teaching course teaching involvement and EMI training.Experience lecturers manifested more negative attitude towards EMI. Elder lecturers had high degree of negative attitude towards higher education EMI.The study should be carried out throughout Malaysia including all lecturers readiness for EMI and for better conclusion of having trained teachers with positive attitude towards readiness for EMI knowledge and for training of EMI.

Reviewer #2: The authors’ examined lecturer readiness to teach via implementation of the English Medium Instruction (EMI) based on self-reporting of lecturer participants. It is unclear what the latter part of the title, “Fantasy or realities?” reference for the purpose of the study. Based on participant responses, the overarching conclusion indicated that lecturers with more knowledge, understanding, and experience in EMI were more accepting of the practice. Thus, following the conclusion, it would be beneficial to know what recommendations the authors have in advancing and facilitating more positive acceptance and readiness of lecturers for EMI at all levels, i.e., what professional development opportunities or programs would be offered? The study was limited to two private higher education institutions and did include representative sample from public institutions of higher education in Malaysia – the conclusion only noted that “current study limitation was the employment of self-reported questionnaire, which potentially produced social desirability response bias.” Please see additional comments below.

Introduction, p.7, 1st sentence of paragraph 1 – Whom do the authors mean when they stated that EMI is “gaining traction, particularly among relevant scholars.” Who are the relevant scholars?

Introduction, p. 9, in the last full paragraph before the Literature Review section, is “academic qualification” interchangeable with “educational qualification”? Also, please clarify what it means by “diversified EMI classrooms” as this phrasing and similar such phrasings were used in the manuscript, for example, diversification of student nationalities, learner pre-knowledge level, learning styles, etc?

While the authors provided a brief literature review of EMI from the perspective of lecturer training, skills, and acceptance, there were no definitions of what EMI training requirements, competencies, and knowledge base entailed. Are the competencies for gaining the skills and knowledge of EMI broad to where the skillset is transferable across all disciplines, is the approach of EMI specific for broad categories of disciplines (e.g., sciences vs liberal arts)? It would be useful to briefly note which disciplines use EMI more frequently.

Methodology, p. 12 – inconsistency between abstract and description in methodology: abstract stated that a survey questionnaire was administered to 227 lecturers, while Methodology section implied distribution of “250 questionnaires in person”; rather than stating in abstract that the questionnaire was completed by 227 lecturers (out of 250 invited participants). The survey completion was noted to be voluntary and anonymous. How was anonymity preserved in receiving completed surveys when the survey was distributed in person?

Table 1 noting the participants’ characteristics is not clear. What does the “Frequency” column reference – the percent of participants in the 4 disciplines noted? Recommend using a more descriptive title representing the data for the table.

Recommend proofreading for consistency in verb tense used throughout (some were present tense while others were past tense) as well as run-on sentences (e.g., Results, p. 15, last sentence before Table 3 and last sentence on p. 19 before Table 8).

While much of the results were reflective of other studies, the postulate “that lecturers without adequate EMI training and preparation would be less…responsive to students’ different learning environments in content knowledge and the English language…” is unclear. How does knowledge of EMI contribute to a lecturers’ responsiveness to students regardless of learning environment, or is this in relation to a lecturer’s capability to respond in the English language?

It would be interesting to learn of a follow-up from this study after an intervention to provide internal faculty/professional development on EMI and resource support using this same survey – how would the attitudes have changed based on participant characteristics? (for both private and public institutions)

6. PLOS authors have the option to publish the peer review history of their article (what does this mean?). If published, this will include your full peer review and any attached files.

Reviewer #1: No

Reviewer #2: No

---

## [Author Response · Author response to Decision Letter 0]

20 Mar 2023

The reviewers have provided us with very insightful comments to improve our article. We gratefully acknowledge their thoroughness and generous feedback.

---

## [Editor Report · Decision Letter 1]

3 Apr 2023

Lecturers' readiness for EMI in Malaysia higher education

PONE-D-22-27417R1

Dear Dr. Yueh Yea Lo,

We’re pleased to inform you that your manuscript has been judged scientifically suitable for publication and will be formally accepted for publication once it meets all outstanding technical requirements.

Kind regards,

Priti Chaudhary, M.S.

Academic Editor

PLOS ONE
---

## [Editor Report · Acceptance letter]

11 Apr 2023

PONE-D-22-27417R1 

Lecturers' readiness for EMI in Malaysia higher education 

Dear Dr. Lo:

I'm pleased to inform you that your manuscript has been deemed suitable for publication in PLOS ONE. Congratulations! Your manuscript is now with our production department. 

Kind regards, 

on behalf of

Dr. Priti Chaudhary 

Academic Editor

PLOS ONE